# Scaling of olfactory antennae of the terrestrial hermit crabs *Coenobita rugosus* and *Coenobita perlatus* during ontogeny

Lindsay D. Waldrop[1,2], Roxanne M. Bantay[2,3] and Quang V. Nguyen[2]

[1] Department of Mathematics, University of North Carolina at Chapel Hill, United States
[2] Department of Integrative Biology, University of California, Berkeley, United States
[3] Department of Biology, San Francisco State University, United States

Corresponding author
Lindsay D. Waldrop,
lwaldrop@email.unc.edu

## ABSTRACT

Although many lineages of terrestrial crustaceans have poor olfactory capabilities, crabs in the family Coenobitidae, including the terrestrial hermit crabs in the genus *Coenobita*, are able to locate food and water using olfactory antennae (antennules) to capture odors from the surrounding air. Terrestrial hermit crabs begin their lives as small marine larvae and must find a suitable place to undergo metamorphosis into a juvenile form, which initiates their transition to land. Juveniles increase in size by more than an order of magnitude to reach adult size. Since odor capture is a process heavily dependent on the size and speed of the antennules and physical properties of the fluid, both the transition from water to air and the large increase in size during ontogeny could impact odor capture. In this study, we examine two species of terrestrial hermit crabs, *Coenobita perlatus* H. Milne-Edwards and *Coenobita rugosus* H. Milne-Edwards, to determine how the antennule morphometrics and kinematics of flicking change in comparison to body size during ontogeny, and how this scaling relationship could impact odor capture by using a simple model of mass transport in flow. Many features of the antennules, including the chemosensory sensilla, scaled allometrically with carapace width and increased slower than expected by isometry, resulting in relatively larger antennules on juvenile animals. Flicking speed scaled as expected with isometry. Our mass-transport model showed that allometric scaling of antennule morphometrics and kinematics leads to thinner boundary layers of attached fluid around the antennule during flicking and higher odorant capture rates as compared to antennules which scaled isometrically. There were no significant differences in morphometric or kinematic measurements between the two species.

## INTRODUCTION

Capturing chemical signals (odors) from the environment is an important task to many animals in both air and water. Crustaceans use the information derived from odors to find food and mates, identify conspecifics, and avoid predators (*Dusenbery, 1992*; *Atema, 1995*; *Zimmer & Butman, 2000*; *Diaz et al., 1999*; *Pardieck et al., 1999*; *Ferner, Smee & Chang, 2005*; *Lecchini et al., 2010*; *Hazlett, 1969*; *Caldwell, 1979*; *Gleeson, 1980*; *Gleeson, 1982*; *Keller,*

*Powell & Weissburg, 2003*; *Gherardi, Tricarico & Atema, 2005*; *Gherardi & Tricarico, 2007*; *Shabani, Kamio & Derby, 2009*; *Skog, 2009*; *Welch et al., 1997*). The olfactory organ of malacostracan crustaceans consists of chemosensory sensillae (aesthetascs) arranged in an array on the lateral flagellum each of their first antennae (antennules) (Fig. 1A). To capture odors, crustaceans move their antennules back and forth through the water in a motion called flicking (Fig. 1A). For aquatic crustaceans, each flick is a sniff, or the capture of a discrete sample of odor-containing water.

Many crustaceans, after moving onto land, have reduced or lost olfactory function in their antennules, few aesthetascs, and are unable to navigate by chemical signals (*Bliss & Mantel, 1968*; *Greenaway, 2003*). Although odor capture by terrestrial crustaceans is less understood as compared to aquatic crustaceans, there is evidence that crabs in the Family Coenobitidae are adept at locating a variety of odor sources in their environment including vegetable matter, detritus, and water (*Thacker, 1996*), as well as other non-traditional food items such as human feces and vomit (*Barnes, 1997*; *Waldrop, 2012*). Coenobitids have robust antennules with arrays of aesthetascs on the lateral flagella (*Ghiradella, Case & Cronshaw, 1968*; *Stensmyr et al., 2005*) and wave their antennules in a flicking behavior similar to aquatic species (*Stensmyr et al., 2005*; *Harzsch & Hansson, 2008*). Coenobitids also have a large olfactory lobe, the site of olfactory information processing, similar in structure to the lobes of insects (*Beltz et al., 2003*).

Despite their high level of terrestrialization, coenobitids are still tied to the ocean for reproduction, releasing marine larvae into the sea to develop (*Greenaway, 2003*). Competent larvae settle in the near-shore shallows and metamorphose into their juvenile, terrestrial form. This metamorphosis is followed by a transition to living exclusively on land (*Brodie, 2002*). The life history of coenobitids presents a unique challenge in terms of odor capture: these crabs must contend with both a change in size and a change in the fluid itself, both of which change the fluid dynamics of odor capture.

## Fluid dynamics of odor capture

Crustaceans flick their antennules to generate convective flows around the sensory structures located on the antennules (*Schmidt & Ache, 1979*; *Moore, Atema & Gerhardt, 1991*; *Moore & Crimaldi, 2004*; *Goldman & Patek, 2002*). When fluid moves across the antennule during flicking, a thin layer of fluid adheres to the surface of the antennule (the 'no-slip' condition); fluid between the layers attached to the antennule and surrounding environment is sheared and creates a velocity gradient from zero to the antennule's speed ($u_\infty$) in a layer of fluid known as a boundary layer (*Denny, 1993*; *Vogel, 1994*).

The relative thickness of the boundary layer depends on the size and speed of the antennule as well as the physical properties of the fluid. In laminar flow, the relative boundary layer thickness ($\delta/L$) is defined as the distance away from a moving object at which the speed of flow is less than 1% of the object's speed ($u_\infty$), and for a flat plate moving through fluid:

$$\frac{\delta}{L} \propto \sqrt{\frac{\nu}{u_\infty L}} \propto Re^{-1/2} \tag{1}$$

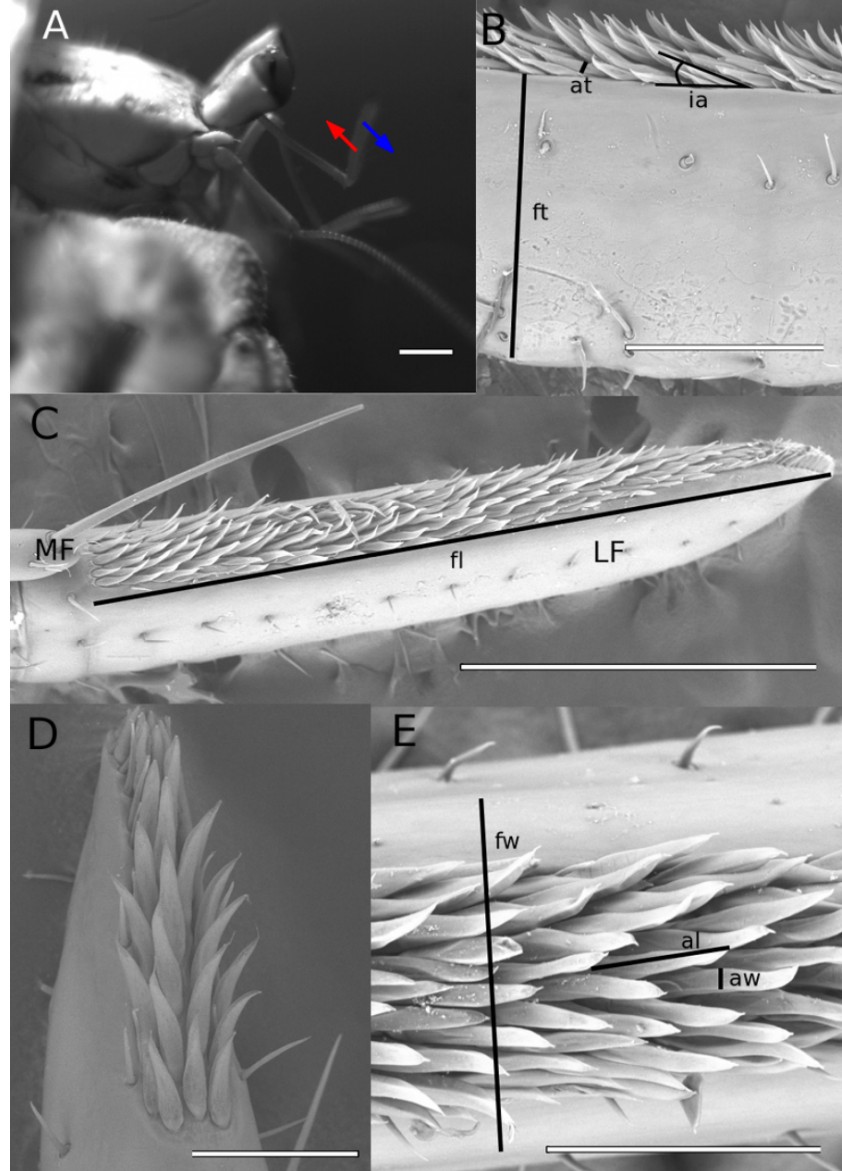

**Figure 1 Scanning electron micrographs of hermit crab anntenules.** (A) Lateral view of the anterior of *C. perlatus*, arrows indicate the direction of antennule flicking of the downstroke (blue) and return stroke (red) (scale: 1 mm). (B–F) Scanning electron micrographs of antennules. (B) Lateral flagellum (LF) of the antennule, lateral view, 999 mm carapace width (CW) animal, with medial flagellum (MF) (scale: 300 μm). (C) 5.43 mm CW (scale: 1 mm). (E) ventral view, 7.51 mm CW (scale: 500 μm). (F) ventral view, 1.2 mm CW (scale: 100 μm). Morphometric measurements: fl, flagellum length; fw, flagellum width; ft, flagellum thickness; al, aesthetasc length; aw, aesthetasc width; at, antennule thickness; ia, insertion angle of aesthetasc.

where the Reynolds number (*Re*) is defined as

$$Re = \frac{u_\infty L}{v} \tag{2}$$

**Peer**J

and relies on the distance from the leading edge of the plate ($L$) (*Vogel, 1994*; *Denny, 1993*). The fluid's kinematic viscosity ($\nu$) in m$^2$ s$^{-1}$ describes how quickly momentum diffuses into a fluid. The flow that induces this boundary layer prevents molecules from directly reaching the olfactory surface of the antennule, but increasing the antennule's speed will decrease the boundary layer thickness. A decrease in boundary layer thickness means a decrease in the distance between new odor-containing fluid and the antennule's surface (*Moore, Atema & Gerhardt, 1991*; *Koehl, 2006*).

Since odor molecules cannot be delivered to the surface of the aesthetascs by fluid currents alone due to the entrained fluid layer, molecular diffusion must transport molecules across the boundary layer (*Stacey, Mead & Koehl, 2002*; *Schuech et al., 2012*). Molecular diffusion is the movement of molecules by a random walk which causes changes in concentration over time. A change in concentration ($C$) with time ($t$) due to molecular diffusion in one dimension ($x$) is described by Fick's law:

$$\frac{\partial C}{\partial t} = D \frac{\partial C}{\partial x} \tag{3}$$

where $D$ is the substance's diffusivity coefficient in m$^2$ s$^{-1}$. The higher the value of $D$, the more quickly molecules diffuse from their original point.

The Peclet number provides a way to determine what mechanism of mass transport (convection or diffusion) will dominate in a particular system. The Peclet number ($Pe = UL/D$) describes the ratio of advective to ($UL$) diffusive ($D$) time scales (*Weissburg, 2000*; *Moore & Crimaldi, 2004*). Advection-dominated systems have $Pe$ above 1, whereas diffusion-dominated systems are below 1. Although the physical process of odor capture is the same in both air and water, the change in fluid causes a major difference between aquatic and terrestrial crab flicking. Crustaceans use a faster motion during the downstroke of flicking to thin the boundary layers between individual aesthetascs so that water can penetrate the space between the aesthetascs, while the slower return stroke serves to trap water between the aesthetascs to give time for odor molecules to diffuse to the sensory surface (*Moore, Atema & Gerhardt, 1991*; *Moore, Scholz & Atema, 1991*; *Koehl, 2006*). For hermit crabs in air, fluid-flow patterns around a dynamically scaled physical model of the antennule during flicking demonstrated that the spaces between the aesthetascs are too small and $Re$ in air are too small to allow for penetration of air within the array (*Waldrop, 2012*). Since the $Pe$ of antennule flicking in air is far below 1 and likely would not enhance odor capture, the odor-capture system of aquatic crustaceans is not observed in adult terrestrial hermit crabs (*Waldrop, 2012*).

## Scaling of odor-capture performance

As a terrestrial hermit crab grows, the increase in size and flicking speed and the change in fluid associated with the water-to-air transition will affect the ability of an animal to capture odor molecules. Odor capture will be affected in two major ways: the boundary layer over which diffusion must transport molecules and the diffusivities in air and water.

The change in concentration away from an object (along $x$) can be thought of as proportional to the velocity gradient of the boundary layer around the antennule, as

presented in *Denny (1993)*:

$$\frac{\partial C}{\partial x} \propto \frac{C_\infty}{u_\infty}\frac{du}{dx}. \tag{4}$$

If we assume the velocity gradient of a flat object in viscous flow as:

$$\frac{du}{dx} \propto \frac{u_\infty Re^{1/2}}{L}. \tag{5}$$

Substituting these relationships into Fick's equation (Eq. (3)), we get an expression that is proportional to mass flux in laminar flow due to convection:

$$\frac{\partial C}{\partial t} \propto \frac{DC_\infty}{L}Re^{1/2} \tag{6}$$

where $C_\infty$ is the concentration of odorant above the boundary layer. Since mass flux due to convection scales as $Re^{1/2}/L$, we can surmise that an animal with an antennule 1/10th the size of an adult antennule would capture about 70% fewer molecules.

However, if the antennules of juvenile crabs were relatively larger as compared to body size, the reduction in *Re*, and therefore odor capture, would be less severe. Allometric scaling has to be investigated in aquatic crustaceans during sniffing (*Mead, Koehl & O'Donnell, 1999*; *Mead & Koehl, 2000*; *Goldman & Koehl, 2001*; *Waldrop, 2012*; *Waldrop, 2013*) and locomotion (*Williams, 1994a*; *Williams, 1994b*). For sniffing crustaceans, allometric scaling is a key feature that allows juvenile stomatopods and crabs to retain the ability to sniff throughout growth.

The physical properties of the fluid also change as a juvenile moves from water to air. These properties affect the movement of odor molecules in the fluid, so a change in fluid will also alter the dynamics of odor capture. Two fluid properties are of interest during this transition: the kinematic viscosity of the fluid ($\nu$) and the molecular diffusivity of a substance in a fluid ($D$). Both kinematic viscosity and molecular diffusivity share the same units, and they can both be thought of as values of diffusivity; molecular diffusivity as the ability of mass to diffuse through the fluid, and kinematic viscosity as the ability of momentum to diffuse through a fluid.

The Schmidt number ($Sc = \nu/D$) describes the relative importance of momentum to mass diffusivity, and previous analyses show that the flux density of molecules is proportional to $Sc^{1/3}$ (*Bird, Stewart & Lightfoot, 1960*; *Denny, 1993*). The kinematic viscosity of air is approximately 15 times higher than water, and the molecular diffusivities of odorant molecules in air are typically several orders of magnitude higher than in water. Moving from water into air would cause a large drop in *Sc*, so flux densities would change drastically during the juvenile's transition from water to air.

## Study objectives

In this study, we characterize the changes in the antennules, aesthetascs, and flicking kinematics during ontogeny of two species of co-occurring terrestrial hermit crabs, *Coenobita rugosus* and *Coenobita perlatus*. From these measurements, we use a simple

model of mass transport to infer how the odorant capture will change during growth for a variety of odorants in both air and water.

# METHODS AND MATERIALS

## Collection and maintenance of animals

Terrestrial hermit crabs of two species, *Coenobita perlatus* H. Milne-Edwards and *Coenobita rugosus* H. Milne-Edwards, were collected on Motu Tiahura, a small island near Moorea, French Polynesia under the auspices of a government scientific permit. Individuals were maintained together in small enclosures outside for approximately 10 days and fed with breadfruit, shrimp, and coconut. After data collection was completed, they were released at the site from where they were collected in accordance with collecting permits.

Carapaces of individuals were photographed and measured for width along the widest point with ImageJ software (*Abramoff, Magalhaes & Ram, 2004*) to the nearest $10^{-4}$ m as an index of body size. Each crab was marked on the dorsal surface of their shell with black permanent marker with unique identifying numbers. Animals ranged in carapace width between 0.80 and 12 mm.

## Kinematics and morphometrics

Videos of antennule flicking by each crab were made for kinematic analysis. Videos were taken inside a well-ventilated room in high relative humidity (85–90%) at an ambient temperature of 80–85 °C. Each animal was placed in a small enclosure (15 cm by 30 cm; 10 cm in height) that limited visual stimulation and ambient wind currents. The enclosure was backed with a black and white grid for determining scale, aligned perpendicularly to the camera; 1 cm$^2$ grid was used for animals over 5 mm in carapace length, and a 1 mm$^2$ grid was used for animals less than 5 mm in carapace width. Animals were allowed to acclimate for five to ten minutes before data collection began. A Phantom Miro high-speed camera was used to record the flicking behavior of antennules at 200 fps.

Flicking events captured on video were analyzed using Graphclick for Mac (Arizona Software, Inc.). Only flicking events that showed the antennule flicking in the plane normal to the camera were selected for analysis. The positions (measured to the nearest $10^{-4}$ m) of the distal end of each lateral flagellum were digitized. The distances traveled between each frame during each stroke were summed over the duration of the stroke, and the summation was divided by the total duration of the stroke to calculate the average speeds for the downstroke and return stroke. Three to five individual flick events were digitized and averaged per individual for each kinematic value.

One antennule from each live animal was excised and preserved on site in 70% ethanol in distilled water and transported to University of California, Berkeley for scanning electron microscopy (SEM). At UCB, undamaged antennules were fixed in 2% glutaraldehyde in 0.1 M sodium cacodylate buffer at pH 7.2 for one hour and then post-fixed in 1 percent osmium tetroxide in 0.1 M sodium cacodylate buffer at pH 7.2. Samples were dehydrated in an alcohol series and dried in a Tousimis AutoSamdri 815 Critical Point Dryer (process described by *Mead, Koehl & O'Donnell (1999)* and *Waldrop (2013)*). Scanning electron

**Table 1 Summary of regressions performed on morphometric measurements.** Regression model used was natural log (measurement) vs. natural log (carapace width in meters) ∗ Species. Widths, lengths, and thicknesses are in meters; angle of aesthetasc with antennule in degrees. Slopes, intercepts, species (*C. rugosus*), and interaction between carapace width and species (*C. rugosus*) are reported as coefficients with their standard errors and *p*-values, which test for significant correlation between the measurement and carapace width. Isometry column gives the expected slope of the regression line if the measurement grow isometrically with carapace width, and the isometry *p*-value, which tests the hypothesis that values grow isometrically ($\beta = \beta_0$).

| Measurement | Slope, $\beta$ | | Intercept | | Species | | Interaction | | Multiple $R^2$ | Isometry | |
|---|---|---|---|---|---|---|---|---|---|---|---|
| | Coef ± SE | *p*-value | Coef ± SE | *p*-value | Coef ± SE | *p*-value | Coef ± SE | *p*-value | | $\beta_0$ | *p*-value |
| **Aesthetasc ...** | | | | | | | | | | | |
| Width | 0.282 ± 0.1 | 0.013[*] | −9.56 ± 0.6 | $2 \times 10^{-16}$[**] | −0.870 ± 0.7 | 0.23 | −0.165 ± 0.1 | 0.21 | 0.24 | 1 | $2 \times 10^{-7}$[**] |
| Length | 0.386 ± 0.1 | 0.004[**] | −7.28 ± 0.5 | $5 \times 10^{-16}$[**] | −0.954 ± 0.7 | 0.15 | −0.1831 ± 0.1 | 0.13 | 0.41 | 1 | $3 \times 10^{-7}$[**] |
| Thickness | 0.267 ± 0.2 | 0.19 | −9.87 ± 1 | $2 \times 10^{-10}$[**] | −1.85 ± 1 | 0.16 | −0.300 ± 0.2 | 0.22 | 0.15 | 1 | $1 \times 10^{-3}$[**] |
| Insertion angle | −0.128 ± 0.2 | 0.50 | 2.07 ± 1 | 0.045[*] | 1.37 ± 1 | 0.26 | 0.2163 ± 0.2 | 0.33 | 0.10 | 0 | 0.49 |
| **Lateral flagellum ....** | | | | | | | | | | | |
| Width | 0.551 ± 0.2 | 0.001[**] | −5.99 ± 0.8 | $2 \times 10^{-7}$[**] | 0.0409 ± 1 | 0.97 | 0.0131 ± 0.2 | 0.94 | 0.73 | 1 | 0.005[**] |
| Length | 0.905 ± 0.2 | $3 \times 10^{-5}$[**] | −1.82 ± 1 | 0.077 | 0.233 ± 1 | 0.86 | 0.00804 ± 0.2 | 0.97 | 0.69 | 1 | 0.61 |
| Thickness | 0.450 ± 0.2 | 0.006[**] | −5.60 ± 0.8 | $1 \times 10^{-7}$[**] | −0.964 ± 1 | 0.37 | −0.197 ± 0.2 | 0.32 | 0.28 | 1 | 0.001[**] |
| Number of aes. | 1.32 ± 0.2 | $6 \times 10^{-7}$[**] | 12.1 ± 1 | $2 \times 10^{-10}$[**] | 0.311 ± 1 | 0.80 | 0.0322 ± 0.2 | 0.89 | 0.91 | 2 | 0.001[**] |

**Notes.**

[*] Indicates significance at level $p < 0.05$.

[**] Indicates significance at level $p < 0.01$.

micrographs were taken with a Hitachi TM-1000 Environmental Scanning Electron Microscope with a 15 kV beam at various magnifications.

Morphometric measurements were taken for one antennule per crab. ImageJ (*Abramoff, Magalhaes & Ram, 2004*) was used to measure (to the nearest $10^{-2}$ µm) morphological features in the scanning electron micrographs. The features of the lateral flagella and aesthetascs that were measured are diagrammed in Figs. 1B, 1C, and 1E. Five aesthetascs from each antennule were haphazardly chosen and measured for width, length and thickness; these measurements were averaged for each individual crab. The width, thickness, and length of the aesthetasc-bearing section of the lateral flagellum, as well as the total number of aesthetascs for each antennule, were measured only once per individual.

## Statistical analysis

Natural logarithms were calculated for each morphometric and kinematic value measured and were graphed against the natural logarithm of carapace width. Regression analysis was completed with the standard statistical package in R (*R Development Core Team, 2011*) to determine if a significant relationship existed between the measurement, carapace width, and species (*Fox & Weisberg, 2011*). The slope, intercept, species effects, and interactive effects for each measurement are reported with standard error and associate *p*-values (Tables 1 and 2). Significance was determined at the level $\alpha = 0.01$.

To determine whether measurements that scale with carapace width scale isometrically or allometrically, expected slopes were determined based on scaling arguments. All

**Table 2 Summary of regressions performed on kinematic measurements.** Regression model used was natural log (measurement) vs. natural log (carapace width in meters) * Species. Speeds are measured in meters per second, durations are in seconds. Slopes, intercepts, species (*C. rugosus*), and interaction between carapace width and species (*C. rugosus*) are reported as coefficients with their standard errors and *p*-values, which test for significant correlation between the measurement and carapace width. Isometry column gives the expected slope of the regression line if the measurement grow isometrically with carapace width, and the isometry *p*-value which tests the hypothesis that values grow isometrically ($\beta = \beta_0$).

| Measurement | Slope, $\beta$ | | Intercept | | Species | | Interaction | | Multiple $R^2$ | Isometry | |
|---|---|---|---|---|---|---|---|---|---|---|---|
| | Coef ± SE | *p*-value | Coef ± SE | *p*-value | Coef ± SE | *p*-value | Coef ± SE | *p*-value | | $\beta_0$ | *p*-value |
| **Downstroke ...** | | | | | | | | | | | |
| Speed | $1.05 \pm 0.1$ | $1 \times 10^{-11}$** | $2.93 \pm 0.6$ | $2 \times 10^{-5}$** | $1.37 \pm 0.9$ | 0.132 | $0.249 \pm 0.2$ | 0.13 | 0.84 | 1 | 0.68 |
| Duration | $0.0694 \pm 0.08$ | 0.41 | $-2.05 \pm 0.5$ | $6 \times 10^{-5}$** | $-1.17 \pm 0.7$ | 0.082 | $-0.199 \pm 0.1$ | 0.01* | 0.09 | – | – |
| **Return stroke ....** | | | | | | | | | | | |
| Speed | $1.32 \pm 0.1$ | $5 \times 10^{-14}$** | $4.66 \pm 0.7$ | $9 \times 10^{-9}$** | $0.625 \pm 0.9$ | 0.50 | $0.116 \pm 0.17$ | 0.49 | 0.87 | 1 | 0.009** |
| Duration | $0.102 \pm 0.09$ | 0.26 | $-2.10 \pm 0.5$ | $1 \times 10^{-4}$** | $-1.41 \pm 0.7$ | 0.05 | $-0.238 \pm 0.1$ | 0.066 | 0.11 | – | – |

Notes.
[*] Indicates significance at level $p < 0.05$.
[**] Indicates significance at level $p < 0.01$.

measurements of length were expected to have slopes equal to 1 ($\beta_0 = 1$), except the number of aesthetascs which had an expected slope of 2 as the number of aesthetascs should be proportional with the aggregate area of the region of the lateral flagellum bearing aesthetascs ($\beta_0 = 2$). Number of aesthetascs was used instead of aggregate aesthetasc area due to the angle at which most micrographs were taken, making area calculations potentially imprecise and inaccurate. Slopes not equal to the expected slope for that measurement indicate allometric growth ($\beta \neq \beta_0$). To test statistical difference between slopes, the measured slope $\beta$ was tested against the expected slope using a *t*-statistic (*Waldrop, 2013*). *p*-values were determined from a student's bimodal *t*-distribution with ($n - 2$) degrees of freedom at $\alpha = 0.01$.

## Model of odor-capture performance

To explore the effects of scaling on the fluid dynamics of odor capture, we calculated the Reynolds number, relative boundary layer thickness, and odorant capture model for two sets of kinematic and morphometric values: (1) the observed allometric relationships identified by the morphometric and kinematics analyses, and (2) an isometric relationship derived from morphometrics and kinematics of the largest animal in the study (carapace width = 0.012 m) and scaled down with a slope equal to 1 to reflect isometry. The largest animal was chosen as the starting point since the fluid dynamics of antennule flicking estimates of absolute odor capture rates are established for adult land hermit crabs in *Waldrop (2012)*.

Reynolds numbers, important for determining the minimum distance required for molecules to diffuse from surrounding air, were calculated using Eq. (2) and the observed, allometric values and calculated isometric values. Antennule width was used as the characteristic length ($L$), stroke speed (downstroke and return stroke separately) was used as velocity relative to the object $u_\infty$, and the value used for the kinematic viscosity of air at

high humidity was $\nu = 8.55 \times 10^{-6}$ m$^2$ s$^{-1}$ (*Denny, 1993*). Reynolds numbers were used to estimate the thickness of the boundary layer surrounding the antennule while flicking, having roughly a relationship proportional to the $Re^{-1/2}/L$.

To investigate how scaling would affect the molecular capture, we constructed a simple model based on the analysis in *Denny (1993)*, which we reconstruct here. Combining Eq. (6) with the idea that the change in concentration is proportional to $Sc^{1/3}$ (*Denny, 1993*; *Bird, Stewart & Lightfoot, 1960*), the resulting equation is an expression of mass flux density due to convection ($J_c$):

$$J_c = \frac{0.32DC_\infty}{L}Re^{1/2}Sc^{1/3}. \tag{7}$$

Since the total mass flux density to an object is due to both diffusion and convection, adding the convective and diffusive terms gives Denny's expression for total mass flux density ($J$):

$$J = \frac{2DC_\infty}{L} + \frac{0.64DC_\infty}{L}Re^{1/2}Sc^{1/3}. \tag{8}$$

When antennule width is used as $L$ and downstroke speed is used as $u_\infty$, Eq. (8) provides an estimate of the density flux of odorant molecules to the antennule. To measure the performance of odorant capture, we multiplied Eq. (8) by the circumference of the antennule to produce a current of odorant molecules per unit antennule length:

$$I = 4\pi DC_\infty + 1.2\pi DC_\infty Re^{1/2}Sc^{1/3}. \tag{9}$$

A range of antennule widths and downstroke speeds were used to calculate $I$ with Eq. (9), based on the carapace widths of animals from which data were collected. Two versions of $I$ were calculated: one based on observed allometry ($I_a$), and one based on isometry ($I_i$). These results are presented as the ratio $I_a/I_i$ that is independent of initial concentration since $C_\infty$ cancels from the Eq. (9) when a ratio is calculated. Since the isometric values were calculated based first on the largest animals, the ratio $I_a/I_i$ will approach 1 at the largest carapace width.

A range of diffusion coefficients was also used to explore the effects of diffusivity on $I_a/I_i$. This range ($D = 1 \times 10^{-12}$ to $1 \times 10^{-5}$ m$^2$ s$^{-1}$) bounds a realistic range of diffusivities in air and water of odorants that have demonstrated physiological responses in *Coenobita clypeatus* (*Krang et al., 2012*).

## RESULTS

### Scaling of morphometric measurements with carapace width

For morphological measurements, 8 *C. perlatus* and 27 *C. rugosus* yielded data, ranging in carapace width from 1.2 to 12 mm. Morphometric measurements of the features of aesthetascs and lateral flagella were graphed against carapace width in Fig. 2. Table 1 is a summary of the regressions performed on these values against carapace width which includes coefficients for slope and intercept as well as multiple $R^2$ for each regression. All

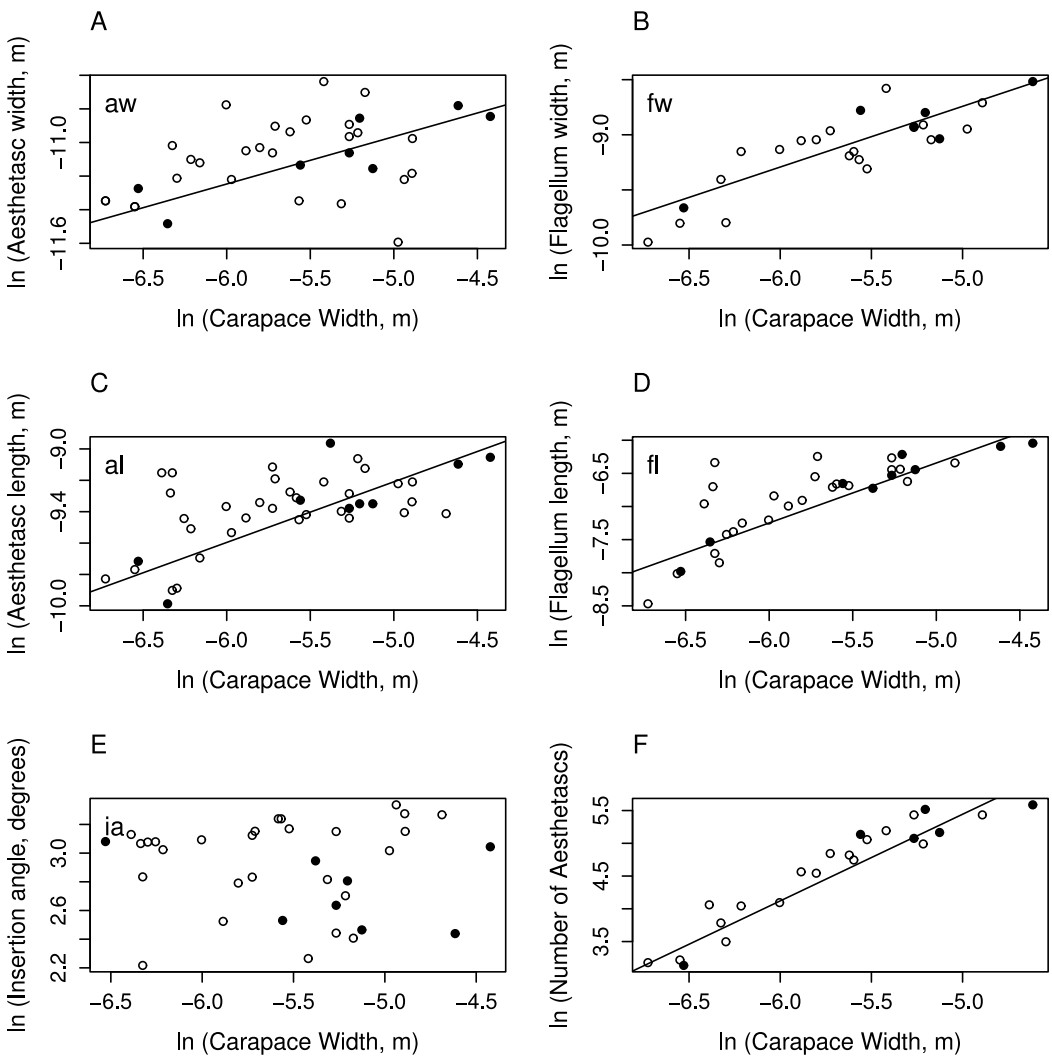

**Figure 2 Morphological features graphed against carapace width.** Natural logarithm of measurements of aesthetasc and flagellum dimensions vs. natural logarithm of carapace width (m). Each measurement corresponds to the dimension illustrated in Fig. 1. Black, filled circles are measurements on *C. perlatus*; white, unfilled circles are measurements on *C. rugosus*. Solid lines represent the significant regression lines. (A) Aesthetasc width (m), (B) flagellum width (m), (C) aesthetasc length (m), (D) flagellum length (m), (E) insertion angle (degrees), and (F) number of aesthetascs in array. There are no significant differences between species in morphometric measurements (see Table 1 for *p*-values).

data reported in Fig. 2 and used to create the statistics in Table 1 are available on the public data repository figshare: http://dx.doi.org/10.6084/m9.figshare.1030320.

Aesthetasc widths ranged between $9.23 \times 10^{-6}$ and $2.4 \times 10^{-5}$ m, and lengths between $4.60 \times 10^{-5}$ and $1.28 \times 10^{-4}$ m. Both aesthetasc length and width scale significantly with carapace width (width: $\beta = 0.282 \pm 0.1$, *dof* $= 31$, $p = 0.013$; length: $\beta = 0.386 \pm 0.1$, *dof* $= 37$, $p = 0.004$), although the regression model seems to explain little of the variation in measurement. Aesthetasc thickness (range: $4.82 \times 10^{-6}$ to $1.69 \times 10^{-5}$ m) and the insertion angle of the aesthetascs (9.18–28.1 degrees) did not vary significantly with

carapace width. The value of observed slope also indicate these features grow with negative allometry, as they are significantly less than the expected slope value of one based on isometric growth (width: $\beta_0 = 1$, $\beta = 0.282 \pm 0.1$, $dof = 31$, $p = 2 \times 10^{-7}$; length: $\beta_0 = 1$, $\beta = 0.386 \pm 0.1$, $dof = 37$, $p = 3 \times 10^{-7}$). Since both slopes are less than one, juvenile hermit crabs have relatively larger aesthetascs compared to adult animals, and aesthetascs grow more slowly during ontogeny than body size.

Several measurements of the aesthetasc-bearing section of the lateral flagellum scaled significantly with carapace width (see Table 1 for $p$-values). Flagellum width and thickness also scale with negative allometry, having slope that are significantly less than the expected slope (width: $\beta_0 = 1$, $\beta = 0.551 \pm 0.1$, $dof = 21$, $p = 0.006$; thickness: $\beta_0 = 1$, $\beta = 0.450 \pm 0.2$, $dof = 32$, $p = 0.001$). These slopes are also reflective of relatively larger antennules on small juvenile hermit crabs which grow more slowly than carapace width. However, the length of the flagellum, ranging from $7.10 \times 10^{-5}$ to $2.3 \times 10^{-3}$ m, had a slope consistent with the expected slope ($\beta_0 = 1$, $\beta = 0.905 \pm 0.2$, $dof = 29$, $p = 0.61$), indicating that the length of the antennule is isometric with body size throughout ontogeny.

The total number of aesthetascs, ranging from 23 to 267, on the lateral flagellum is proportional to the area of the array and should scale with carapace width as area scales with length; the expected slope of this relationship is two. The number of aesthetascs does scale with carapace width ($\beta = 1.32 \pm 0.2$, $dof = 19$, $p = 6 \times 10^{-7}$) but significantly less than the value expected from isometry ($\beta_0 = 2$, $\beta = 1.32 \pm 0.2$, $dof = 19$, $p = 0.009$). This result indicates that adult animals have relatively fewer aesthetascs in their arrays than small juveniles.

## Scaling of kinematic measurements with carapace width

For kinematic measurements, 16 *C. perlatus* and 29 *C. rugosus* yielded data, ranging in carapace width from 0.80 to 12 mm. This range differs slightly from the range of carapace widths for morphometrics because their were no flicking events for the smallest animal, 0.80 mm, that met inclusion criteria for analysis. Kinematic measurements of antennule movement were graphed against carapace width (Fig. 3). All data reported in Fig. 3 and used to create the statistics in Table 2 are available on the public data repository figshare: 10.6084/m9.figshare.1030320.

Downstroke speeds ranged between $7.39 \times 10^{-3}$ m s$^{-1}$ and $1.49 \times 10^{-1}$ m s$^{-1}$ and durations ranged from $5.88 \times 10^{-2}$ to $1.55 \times 10^{-1}$ s. Downstroke scaled significantly with carapace width ($\beta = 1.05 \pm 0.1$, $dof = 41$, $p = 1 \times 10^{-11}$). The observed slope did not differ significantly from the expected slope ($\beta_0 = 1$, $\beta = 1.05 \pm 0.1$, $dof = 41$, $p = 0.68$), indicating that downstroke speed scaled with isometry. Downstroke duration, however, was not significantly correlated with carapace width ($\beta = 0.0694 \pm 0.08$, $dof = 39$, $p = 0.41$).

Return stroke speeds ranged from $7.55 \times 10^{-3}$ to $43.72 \times 10^{-1}$ m s$^{-1}$ and durations ranged between $3.67 \times 10^{-2}$ and $1.53 \times 10^{-1}$ s. Like the downstroke speeds, return stroke speeds increased significantly with increasing carapace width ($\beta = 1.32 \pm 0.1$, $dof = 41$, $p = 5 \times 10^{-14}$), but increase faster than predicted ($\beta_0 = 1$, $\beta = 1.32 \pm 0.1$, $dof = 41$,

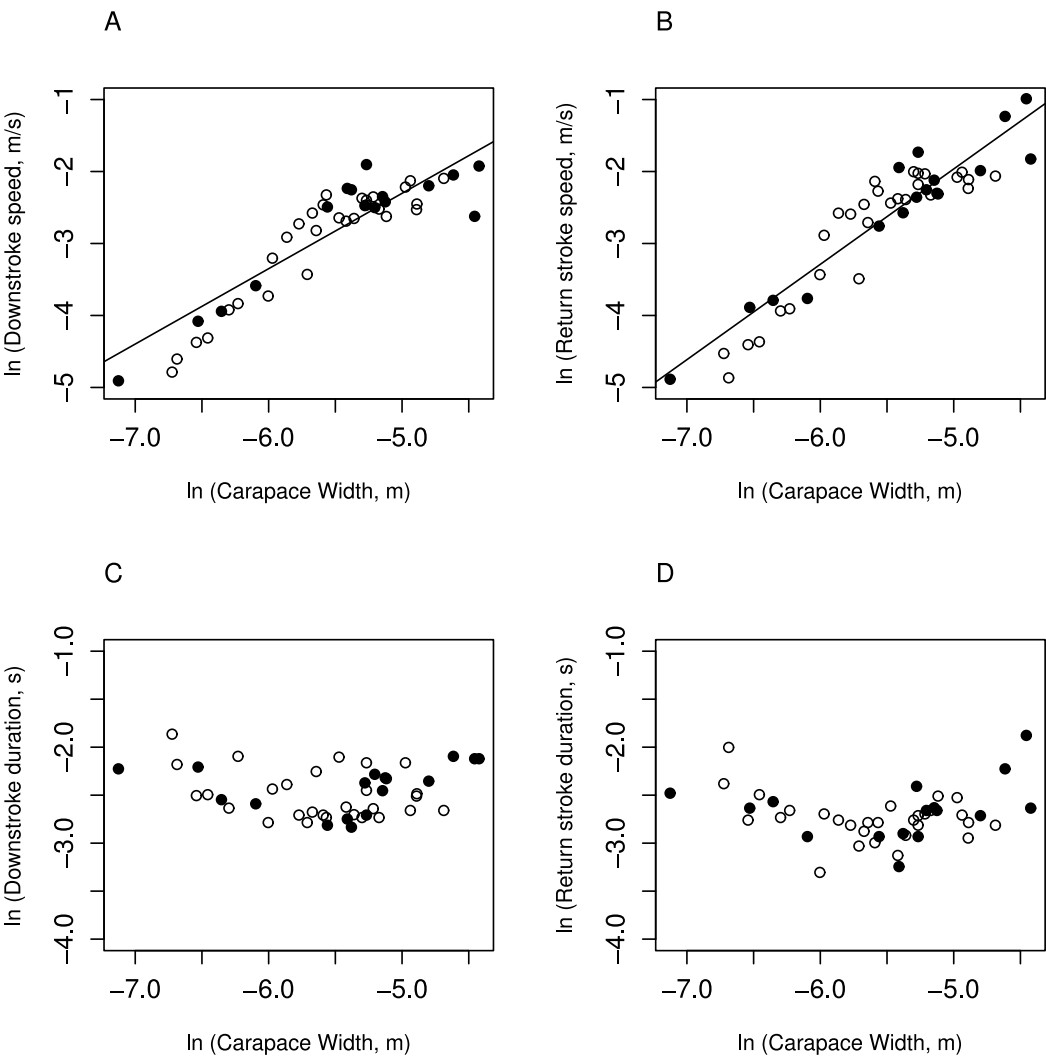

**Figure 3 Kinematic measurements graphed against carapace width.** Natural logarithm of kinematic measurements versus natural logarithm of carapace width (m). Black, filled circles are measurements on *C. perlatus*; white, unfilled circles are measurements on *C. rugosus*. Solid lines represent the significant regression lines. (A) Downstroke speed (m s$^{-1}$); (B) return stroke speed (m s$^{-1}$); (C) downstroke duration (s); and (D) return stroke duration (s). There are no significant differences between species in kinematic measurements (see Table 2 for *p*-values).

$p = 0.009$). Return stroke durations were not significantly correlated with carapace width ($\beta = 0.102 \pm 0.09$, $dof = 41$, $p = 0.26$).

## Effects of scaling on molecule capture

Reynolds numbers for the downstroke (blue lines) and return stroke (red lines) are plotted against carapace width in Fig. 4A. The solid lines represent *Re* based on allometric measurements and dashed lines are *Re* calculated based on isometry. *Re* for both stroke increase during ontogeny and for observed allometry and isometry. For the allometric condition, downstroke *Re* range between 0.060 and 4.6 and between 0.050 and 7.9 for

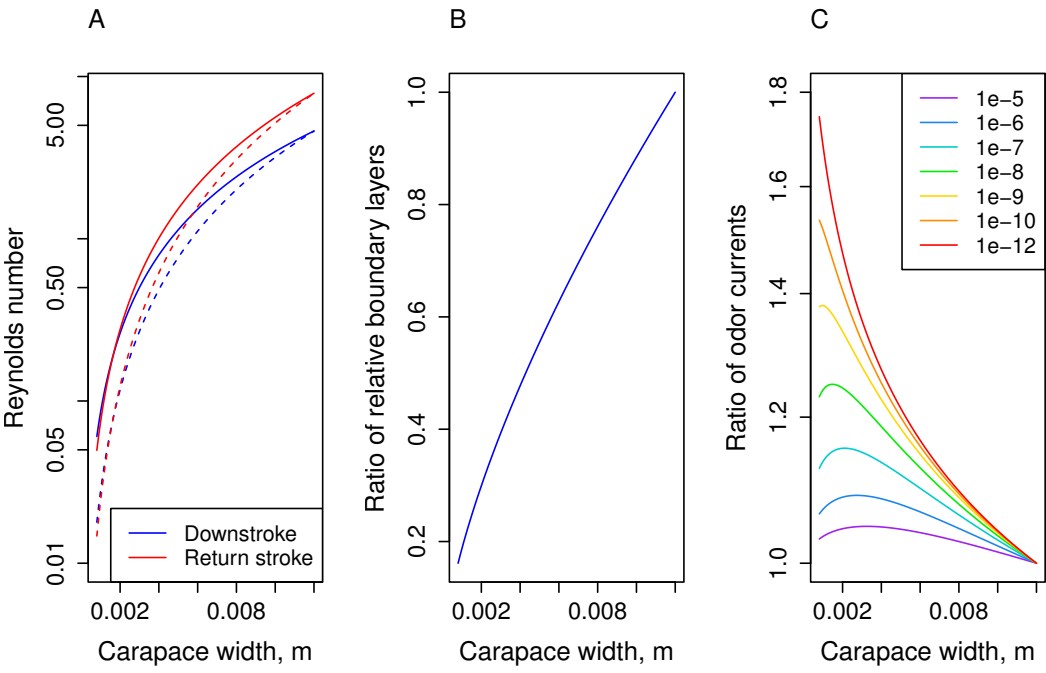

**Figure 4 Calculated values based on regression lines versus carapace width.** (A) Reynolds numbers of antennule flicking ($L$ = flagellum widths) on a logarithmic scale, solid lines are based on allometric relationships and dashed lines are based on isometry; (B) ratio of boundary layer thicknesses predicted by allometry to isometry; and (C) ratio of odor current per unit antennule length predicted by allometry to isometry ($I_a/I_i$) using various values of $D$ (m$^2$ s$^{-1}$). Values converge to 1 at the largest carapace widths. Higher diffusion coefficients ($1 \times 10^{-7}$ to $1 \times 10^{-5}$ m$^2$ s$^{-1}$) represent molecules in air, lower coefficients ($1 \times 10^{-12}$ to $1 \times 10^{-8}$ m$^2$ s$^{-1}$) represent molecules in water.

the return stroke. The *Re* of the isometric condition start lower for smaller animals (downstroke: 0.018, return stroke: 0.015) while ending at the same values for larger animals, resulting in a larger lifetime change.

The relative boundary layer thicknesses for the downstroke were used to create a proxy for the thickness of the attached fluid layer on the antennule during movement. Ratios were calculated by dividing the thickness based on observed allometry with the thickness based on isometry for each carapace width to identify relative effects of scaling; these ratios is plotted against carapace width in Fig. 4B. For the smallest animals (carapace width around 0.8 mm), the ratio is 0.16, indicating that the boundary layer around an allometrically scaled antennule is 84% thinner than the layer around an isometrically scaled antennule. The ratios approach 1 during growth.

The ratios of odorant currents, $I_a/I_i$, for various diffusion coefficients are graphed against carapace width in Fig. 4C. Because adult values were used to create the scaled-down, isometric juveniles, ratios above 1 indicate that antennules that scale with negative allometry capture more odorant molecules than hypothetical antennules that scale isometrically. The magnitude of this effect depends on both the value of the diffusion coefficient and the carapace width of the animal. The ratio is always highest for smaller animals, but the maximum difference varies between a 5–15% increase for higher diffusion

coefficients ($1 \times 10^{-7}$ to $1 \times 10^{-5}$ m$^2$ s$^{-1}$) and up to 80% increase for lower diffusion coefficients ($1 \times 10^{-12}$ to $1 \times 10^{-8}$ m$^2$ s$^{-1}$).

## Effect of species on morphometric and kinematic measurements

Species was used as a secondary predictor for the regressions of morphometric and kinematic measurements versus carapace width. Tables 1 and 2 report the coefficients and $p$-values associated with the effect of species, as well as the interaction between species and carapace width for each regression model. There was no significant effect of species or interactive effect between species and carapace width for any morphometric measurement examined. There were two slight effects of species in the duration of each stroke where $p$-values were between 0.05 and 0.01, indicating a possible difference in the duration of each stroke between species but not the speed of the antennule movement.

# DISCUSSION

## Scaling of the olfactory antennae and flicking kinematics of terrestrial hermit crabs

Many crustaceans rely on flicking antennules with arrays of chemosensory aesthetascs in order to capture odors in their environments. These animals increase in size over an order of magnitude during growth, and since the fluid dynamics of odor capture rely heavily on the size and speed of the antennules during flicking, the animals' ability to capture odors may change during ontogeny. However, many aquatic crustaceans grow with negative allometry, which reduces the magnitude of size change (*Mead, Koehl & O'Donnell, 1999*; *Waldrop, 2013*) and retains the function of the antennule, that of capturing odors, throughout ontogeny (*Mead & Koehl, 2000*; *Waldrop, 2012*).

Although terrestrial hermit crabs do not seem to rely on the same mechanism of sniffing as aquatic crustaceans such as crabs (*Waldrop, 2012*), many features of their antennules and aesthetasc arrays scale with the same negative allometry as other aquatic crustaceans during ontogeny. Small juveniles have relatively larger antennules and aesthetascs than adult hermit crabs, which results in much higher *Re* and thinner boundary layers than would be expected from isometry. Indeed, the regression coefficients of aesthetasc width and length as well as lateral-flagellum width with carapace width are nearly indistinguishable between hermit crabs and the shore crab *Hemigrapsus oregonensis* (Decapoda: Brachyura) reported in *Waldrop (2013)*. The allometric growth patterns may reflect the phylogenetic history of terrestrial crabs, functional constraints on the chemosensory structure itself (minimum size of an aesthetasc), a result of selective pressures due to differential performance in odor capture, or any combination of the above.

Where the ontogenetic morphometrics of aquatic and terrestrial crabs are strikingly similar, the kinematics of antennule flicking are as strikingly different. Both the downstroke and return stroke of *H. oregonensis* display negative allometry during growth, which results in relatively faster stroke speeds and maintains the *Re* of both strokes in the range important for discrete odor capture (*Waldrop, 2012*; *Waldrop, 2013*). As a sharp contrast to aquatic crabs, the speed of the return stroke scales with positive allometry and the speed of the downstroke is isometric for terrestrial hermit crabs. The return stroke is relatively faster

for larger animals than smaller animals. The combined scaling of antennule morphology and kinematics results in no difference in the $Re$ between the strokes for the smallest juveniles, with the difference growing after surpassing $Re = 1$. Terrestrial hermit crabs do not discretely capture odors like aquatic crustaceans, and the small difference that develops after leaving Stokes flow regime ($Re > 1$) maintains the speed profile on the side of the antennule where aesthetascs are present (*Waldrop, 2012*). This difference between strokes is not necessary for small juveniles in Stokes flow, where flow is reversible and time independent.

### Species and habitat

Morphometric and kinematic data were collected from two species of the genus *Coenobita* which co-occur on tropical Indo-Pacific islands such as in French Polynesia. In fact, the antennules of these species are difficult to tell apart when removed from the animals except for their color (*C. perlatus* antennules are red-orange whereas *C. rugosus* are brown). No metric measured yielded a significant difference between the two species. This result could be due to phylogenetic history, it is unclear how long ago the two species diverged evolutionarily. There may also be selective pressures on performance to maintain the morphology and kinematics of the antennules.

Coenobita species often prefer different terrestrial habitats (*Barnes, 1997*), and we observed *C. rugosus* more often further from shore in wooded inland and *C. perlatus* more often on the near-water shore and hypothesized that this difference in habitat may lead to differences in antennule morphology and flicking kinematics since forest present a different aerodynamic environment for olfactory navigation. However, our results do not indicate such a difference, and when wind data were gathered and analyzed from both habitats where crabs were collected, no difference in wind conditions were observed (*Waldrop, 2014*).

### Effects of ontogenetic scaling on odor-capture performance

In this study, we used a previous analysis (*Denny, 1993*) of mass transport to an object over flow to construct a simple model of odorant capture to infer the effects that growth would have by comparing cases of negative allometric growth and isometric growth. The model suggests that there could be modest increase in odorant capture due to scaling antennule features with negative allometry for all odorant molecules as compared to odorant capture by isometrically scaled antennules. Since the size of an object heavily influences the thickness of the boundary layer (Eqs. (1) and (2)), and thus the distance diffusion must move molecules before capture, the relatively larger antennules of small juvenile crabs due to allometrically growth result in higher molecule capture compared to isometrically scaled antennules.

For small juvenile crabs in air, odorant molecules have diffusivities generally between $D = 1 \times 10^{-7}$ to $1 \times 10^{-5}$ m$^2$ s$^{-1}$ (Fig. 4C, purple, dark blue, and light blue). For these values of diffusivity, there was a modest increase of between 5 and 15% for the observed allometric antennules as compared to isometric antennules. This result suggests that the negative allometry observed in hermit crab antennules could increase the odorant flux to the juvenile's antennule compared to hypothetical antennules that scale isometrically.

The effects of allometry are greater as $D$ decreases, and higher values of $D$ are typical of larger molecules. Odorant molecules are typically volatile organic compounds (dark blue to green in Fig. 4C) which sometimes have lower diffusivities than smaller molecules such as water (purple in Fig. 4C). The model's results suggest that allometric growth results in disproportionately greater delivery of large molecules, while providing little benefit to smaller molecules. If the reverse situation is considered, where water was being lost by diffusion away from the antennule to dry air, extra water would not be lost due to the larger antennules produced by allometry. Recent work by *Krang et al. (2012)* found that exposure to dry air inhibited the ability of hermit crabs to detect odorants, and juvenile hermit crabs would have a heavier burden of water loss with which to contend due to their higher ratio of surface area to volume. Although, the differential effects of scaling would be affected also by the magnitude gradients of both water molecules (humidity) and odorant molecules between aesthetasc and air, and also complicated significantly by the effects of surface tension on water evaporation, which are not a part of this analysis.

### Implications of allometric scaling on settlement of larvae

Olfaction is an important aspect of the transition between sea and land that terrestrial hermit crabs undergo during their lifetimes. Olfactory cues are important for predatory avoidance (*Diaz et al., 1999*) and possibly used to find suitable places to settle (*Welch et al., 1997*). There is evidence that both larvae and juvenile *Coenobita* sp. have well-developed antennules with aesthetascs and olfactory bulbs capable of processing olfactory information, indicating that competent larvae could attempt to find chemical cues in water using their antennules and aesthetasc arrays.

We were able to use this model to analyze the effect on the capture of different odorant molecules in both air and water by varying the diffusion coefficient $D$. While the kinematic viscosities ($\nu$) of air and water differ by 10-fold, the diffusion coefficient for similarly sized molecules is 10,000 times lower in water than air. In our model, using diffusion coefficients in Fig. 4C represented by the warmer colors ($D = 1 \times 10^{-12}$ to $1 \times 10^{-9}$ m s$^{-1}$) that cover a broad range of molecules diffusing in water show a marked improvement for allometrically scaled antennules. Up to 75% more odorant molecules are captured for the smallest juvenile antennules as compared to isometrically scaled antennules for these higher values of $D$. There is a large, non-linear increase in the difference between odorant capture by the observed allometrically scaled and isometrically scale antennules which increases as diffusivities decrease. It is likely that the increase in odorant-delivery due to negative allometry is more apparent to settling post-larvae and juvenile animals that have yet to transition to land.

### Limitations of the model of odor-capture performance

Although this model reveals interesting trends between the negative allometry in the growth of odor-capture structures and odorant capture, it should be considered by the reader as a rough estimate of these effects. Our model suffers from many limitations including using those of geometry (using a flat plate instead of a cylinder in 2D flow to estimate boundary-layer thickness), assuming that odorant molecules can be captured

**PeerJ** ______________________________________________

along the entire circumference of the antennule when capture is restricted to the aesthetascs in the animal, and the absence of odorant concentration differences. A mathematically rigorous treatment of a more realistic geometry is needed to validate these trends.

## ACKNOWLEDGEMENTS

Special thanks to M Wright for support in the field; F Murphy, H Murphy, and the UC Berkeley Gump Field Station for field accommodations; UC Berkeley's CiBER center for equipment; MAR Koehl for support and guidance; L Miller, S Khatri, J Prairie, and D Evangelista for assistance with data analysis and manuscript editing; G Min and the UC Berkeley Electron Microscopy Lab for assistance with microscopy; and M Weissburg and an anonymous reviewer for comments that significantly improved the manuscript.

### Funding

This work was supported by the Virginia and Robert Gill Chair and a MacArthur Foundation Fellowship (to M Koehl); the Reskteco and Gray Fellowships from the University of California, Berkeley Department of Integrative Biology to L Waldrop; Sigma-Xi Grant-in-Aid of Research to the L Waldrop; and by a Traineeship to L Waldrop (National Science Foundation Integrative Graduate Education and Research Traineeship [DGE-0903711] to R Full, M Koehl, R Dudley, and R Fearing); and by a National Science Foundation Research and Training Grant 5-54990-2311 (to R McLaughlin, R Camassa, L Miller, G Forest, and P Mucha). The funders had no role in study design, data collection and analysis, decision to publish, or preparation of the manuscript.

### Grant Disclosures

The following grant information was disclosed by the authors:
MacArthur Foundation Fellowship.
University of California, Berkeley Department of Integrative Biology.
Sigma-Xi Grant-in-Aid of Research.
National Science Foundation Integrative Graduate Education and Research Traineeship: DGE-0903711.
National Science Foundation Research and Training: 5-54990-2311.

### Competing Interests

The authors declare there are no competing interests.

### Author Contributions

- Lindsay D. Waldrop conceived and designed the experiments, performed the experiments, analyzed the data, wrote the paper, prepared figures and/or tables, reviewed drafts of the paper.
- Roxanne M. Bantay and Quang V. Nguyen performed the experiments, reviewed drafts of the paper.

**Peer**J

## Field Study Permissions

The following information was supplied relating to field study approvals (i.e., approving body and any reference numbers):

Approval letter for this study was obtained by L Waldrop from the government of French Polynesia prior to undertaking research.

## Data Deposition

The following information was supplied regarding the deposition of related data:

Figshare DOI: http://dx.doi.org/10.6084/m9.figshare.1030320.

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
