# Peer review of "Scaling of olfactory antennae of the terrestrial hermit crabs Coenobita rugosus and Coenobita perlatus during ontogeny"

_PeerJ, doi:10.7717/peerj.535_

## Round 0.1 · original submission · Major Revisions

· Academic Editor

Major Revisions

Overall both referees are enthusiastic about the manuscript, while both also point out several areas for improvement prior to publication. In particular the second referee has made a number of specific comments that are likely to require re-review, the most important of which are 1) the concern over the discussion of negative allometry where the referee points out important comparisons and other literature that the authors ignore, and 2) the question of whether Eqn 9 is appropriate or defensible. Because these points will likely alter the manuscript in a substantial way, and the results may be altered by the quatitative reassessment of the odorant flux using Eqn 9, I consdier this a major revision.

Reviewer 1 ·

Basic reporting

No comments.

The submission is clearly written, unambiguous, and conforms to professional standard. It includes sufficient introduction and background, and relevant prior literature regarding crustacean olfaction is appropriately referenced. The submission follows the PeerJ template; figures are appropriate and the submission is self-contained.

Experimental design

In order to replicate kinematics measurements, the methods could provide more detail about how the kinematics were filmed, including calibration details and how movements were accepted or rejected for digitizing (the movement appears to be planar?) The methods for morphometrics appear to be adequately described to replicate.

Otherwise, the submission describes original primary research in biology; the research question is clearly defined as the changes in the olfactory aparatus during ontogeny and during shifts from water to air. The study is rigorous and was conducted to technical standards typical for biomechanical work on olfaction. Research conformed to ethical/legal standards.

Validity of the findings

Data appear robust and statistically sound. Raw data is provided on the plots and the statistical analyses (primarily regression on ln-reduced major axes) are provided in the text.

As a control, it would have been nice to see the scaling of some other body part near the olfactory apparatus but perhaps not participating in olfaction. I'm not sure if the authors have such data; it is nice to have but not necessary. Perhaps if olfaction is driving this particular scaling, the same patterns may not be seen in some other body part.

The plots (and tables) themselves are given in ln coordinates (e.g. -7.0, -6.0, -5.0 ln(m) for carapace width). This makes it hard to read the graphs; it would be nice if they could be in log coordinates but with the tics given in plain m. This would make the data more useful and more easily compared to other datasets. Figure 3 may be improved if the caption would note that there are not species differences in the kinematics (P=xxxx).

Much use is made of scaling relationships for boundary layers on flat plates (typical for work coming from Koehl Lab); however the geometry pictured in figure 1 is clearly not a flat plate. In other work, the aesthetasc array is presented as an array of tubes, for example. (I'm not sure there is an easy closed-form solution for the boundary layer of a cylinder at very low Re). The authors should comment on the applicability of flat plate boundary layers to these alternate geometries, especially in cases where the presumed boundary layer is large compared to the geometry of the hairs or antennules. Perhaps one way to do this might be to provide some bounds or expected error bars on the projected odor currents (Figure 4c)? It might be nice to know if 2D and 3D effects would change this picture by 2% or 2000%? On the other hand, this may be more involved than is necessary for the first-order scaling estimates here - if this is what the authors intend then perhaps a few disclaimers or qualifiers may help.

At the very last sentence (unfortunate position!) the author cites her own unpublished data regarding wind conditions. Does she mean for this to be here? If there were no differences in wind condition, perhaps this could be included here; alternatively if it is in some other publication or thesis (Waldrop, 2012?) she should consider providing reference to that, or posting the data where she can get a DOI and "credit" for it?

Additional comments

(around line 50): Diffusion arises from molecular movement? (There are other examples of diffusion that are not molecular movements; and random walks become diffusion only in the limit of continuum hypotheses...)

(around line 130): Seems odd Waldrop 2013 is cited in methods for how to do a t-test? Also, R can be "cited" (Type citation() in R to see how).

(Figure 2): The table and the text have the information, but for the lazy, the authors may wish to consider putting suggestive lines, stars, or whatever on Figure 2 where the scaling differs from isometry. This is a key finding of the paper, so it would be nice to highlight it in the figure so that it grabs people's attention.

---

## Round 0.2 · Minor Revisions

· Academic Editor

Minor Revisions

Both referees have now completed their review and agree that the paper is nearly ready for publication pending some minor fevisions for language and grammar. Each has attached a document with suggested edits for your use, so please follow their suggestions in revising the text for final acceptance. There appears to be some confusion regarding your conclusions presented in lines 324 to 330 that need to be addressed and clarified in the text so that future readers do not share the same confusion as the current referees.

I do not anticipate sending the manuscript out for additional review and expect that these revisions should be relatively straight forward. If you have any questions or need any assistance with your final revisions, please let me know. I look forward to seeing your revised manuscript.

Reviewer 1 ·

Basic reporting

No further comments, previous comments addressed adequately.

Experimental design

No further comments, previous comments addressed adequately.

Validity of the findings

No further comments, previous comments addressed adequately

Additional comments

1. Regarding ln-ln coordinates in Figure 2: I had suggested plotting figures 2 and 3 so that normal units are visible on the axes; it is easy to do in R (see attached done using the data from Figshare and table 1). I generally feel this is preferable because it gives a reader a sense of what the measurements were (2mm to 1 cm carapace width, for example, rather than ln(whatever); and this is how allometric plots are done in some textbooks (Schmidt Nielsen, Vogel, etc). However, in further review, I see that other sources (for example, a review article by LaBarbera; others when doing quick Google search) will plot the log transformed axes in log units, presumably to allow someone to check that the m and b regression parameters are good. My personal preference is the first, but the second appears to be perfectly valid and the authors have gone with it.

2. There are some mistakes in the caption for Fig 1. It refers to panel F which is not present. Also it mentions a 999-mm carapace width animal which would be quite impressive indeed. I suggest the authors tweak this slightly and trust them to do so during proofing (does PeerJ do that?) without needing further review.

3. They may also consider putting the same abbreviation keys ("aw", "al", etc) on the corresponding plots of Fig 2 to allow readers to quickly connect the two without digging through the captions (this latter bit is optional). The new added sentences in Fig 2 and 3 might sound a little less awkward if reworded to say "There are no significant differences between species in... measurements" (switch singular to plural).

Annotated reviews are not available for download in order to protect the identity of reviewers who chose to remain anonymous.

Reviewer 2 ·

Basic reporting

Very sound

Experimental design

technically proficient and appropriate

Validity of the findings

interpreted and explained accurately and concisely

Additional comments

A couple of minor language and style glitches, but otherwise very well done.

Annotated reviews are not available for download in order to protect the identity of reviewers who chose to remain anonymous.

---

## Round 0.3 · accepted · Accept

· Academic Editor

Accept

Your revisions have cleared up the last remaining issues with the manuscript, and I am happy to accpet your paper for publication.